In-situ incubation of a coral patch for community-scale assessment of metabolic and chemical processes on a reef slope

http://orcid.org/0000-0003-2001-8710 van Heuven Steven M.A.C. 1
Webb Alice E. 1
de Bakker Didier M. 2 3
Meesters Erik 3
van Duyl Fleur C. 2
http://orcid.org/0000-0002-7256-2243 Reichart Gert-Jan 1 4
de Nooijer Lennart J. 1 ldenooijer@nioz.nl
1 Department of Ocean Sciences, NIOZ Royal Netherlands Institute for Sea Research, and Utrecht University , Den Hoorn, Noord-Holland , The Netherlands
2 Department of Marine Microbiology, NIOZ Royal Netherlands Institute for Sea Research, and Utrecht University , Den Hoorn, Noord-Holland , The Netherlands
3 Wageningen Marine Research, Wageningen University and Research , Den Helder, Noord-Holland , The Netherlands
4 Department of Earth Sciences, Utrecht University , Utrecht, Utrecht , The Netherlands
Nelson Craig
Electronic publication date: 2018 Dec 3
Publication date: 2018
Volume: 6
Electronic Location ID: e5966
Received 2018 Apr 4; Accepted 2018 Oct 18
Copyright: © 2018 van Heuven et al.
Copyright year: 2018
Copyright holder: van Heuven et al.
License: This is an open access article distributed under the terms of the Creative Commons Attribution License, which permits unrestricted use, distribution, reproduction and adaptation in any medium and for any purpose provided that it is properly attributed. For attribution, the original author(s), title, publication source (PeerJ) and either DOI or URL of the article must be cited.
License URL: https://creativecommons.org/licenses/by/4.0/

Keywords: Coral reef, Incubation, Alkalinity anomaly

Funding: The Netherlands Organisation for Scientific Research (NWO) 858.14.020 The Ministry of Agriculture, Nature and Food Quality, program Caribbean Netherlands BO-43-021.04 The research was funded by the Netherlands Organisation for Scientific Research (NWO), grant 858.14.020. Erik Meesters was financially supported by the Ministry of Agriculture, Nature and Food Quality, program Caribbean Netherlands (BO-43-021.04). The funders had no role in study design, data collection and analysis, decision to publish, or preparation of the manuscript.

==============================
Anthropogenic pressures threaten the health of coral reefs globally. Some of these pressures directly affect coral functioning, while others are indirect, for example by promoting the capacity of bioeroders to dissolve coral aragonite. To assess the coral reef status, it is necessary to validate community-scale measurements of metabolic and geochemical processes in the field, by determining fluxes from enclosed coral reef patches. Here, we investigate diurnal trends of carbonate chemistry, dissolved organic carbon, oxygen, and nutrients on a 20 m deep coral reef patch offshore from the island of Saba, Dutch Caribbean by means of tent incubations. The obtained trends are related to benthic carbon fluxes by quantifying net community calcification (NCC) and net community production (NCP). The relatively strong currents and swell-induced near-bottom surge at this location caused minor seawater exchange between the incubated reef and ambient water. Employing a compensating interpretive model, the exchange is used to our advantage as it maintains reasonably ventilated conditions, which conceivably prevents metabolic arrest during incubation periods of multiple hours. No diurnal trends in carbonate chemistry were detected and all net diurnal rates of production were strongly skewed towards respiration suggesting net heterotrophy in all incubations. The NCC inferred from our incubations ranges from −0.2 to 1.4 mmol CaCO3 m−2 h−1 (−0.2 to 1.2 kg CaCO3 m−2 year−1) and NCP varies from −9 to −21.7 mmol m−2 h−1 (net respiration). When comparing to the consensus-based ReefBudget approach, the estimated NCC rate for the incubated full planar area (0.36 kg CaCO3 m−2 year−1) was lower, but still within range of the different NCC inferred from our incubations. Field trials indicate that the tent-based incubation as presented here, coupled with an appropriate interpretive model, is an effective tool to investigate, in situ, the state of coral reef patches even when located in a relatively hydrodynamic environment.

Introduction

The functionality of many reef systems is intrinsically linked to their structural habitat complexity (Newman et al., 2006; Graham & Nash, 2013; Kennedy et al., 2013). On tropical coral reefs, the three-dimensional habitat relies primarily on the ability of corals to deposit large quantities of calcium carbonate. Over recent decades, corals reefs have been under threat at a global scale by a large number of anthropogenic impacts such as ocean warming, overfishing, eutrophication, and ocean acidification (Hoegh-Guldberg, 1999; Gardner et al., 2003; Hoegh-Guldberg et al., 2007; De’ath et al., 2012; Anthony et al., 2008; Baker, Glynn & Riegl, 2008).

The consequential decline in coral cover and the reduction in historically dominant framework building coral species has already resulted in a substantial loss of 3D-complexity on many tropical reefs (Edinger & Risk, 2000; Alvarez-Filip et al., 2011a; Perry et al., 2015; De Bakker et al., 2016; Hughes, 1994; Hughes et al., 2007).

The impact of individual aspects of environmental change on coral reef health has been assessed in a number of laboratory experiments (Gilmour, 1999; Burkepile & Hay, 2009). However, in situ community-scale measurements of metabolic and geochemical processes would enable characterization of total reef metabolism. Net community calcification (NCC) is considered to reflect the overall response of the community to environmental change and is therefore monitored as a proxy for coral reefs’ status (Gattuso et al., 1993; Kleypas et al., 1999; Edinger et al., 2000). Field validation of coral accretion/decline is required to test whether observed experimental responses can be translated to whole ecosystems and in situ conditions. Coral reef decline or community compositional change can be estimated qualitatively from visual inspection of the same site over time (Aronson & Precht, 1997), or by digital comparison of photographs taken at intervals (Porter & Meier, 1992; Coles & Brown, 2007; De Bakker et al., 2016). The disadvantage of such visual assessments, however, is that results are confined to areas that have been visited previously and are not quantitative with respect to NCC. Still, the latter may be estimated from visual inspections using typical, species-specific calcification rates (Perry, Spencer & Kench, 2008). And although demonstrated to have fair accuracy (Porter & Meier, 1992; Alevizon & Porter, 2015; Chow et al., 2016), “carbonate budgeting” estimates do not allow estimating seasonal variability of NCC (Courtney et al., 2016), and are inherently insensitive to rapid environmental change. The same goes for elevation-change analyses using coral cores as alone do not relate alteration in seafloor structure to cause (Hubbard, Miller & Scaturo, 1990; Yates et al., 2017). Furthermore, the integrated effects due to organismal interactions cannot be assessed with such an approach. For example, ocean acidification may reduce coral NCC (Andersson & Gledhill, 2013) and at the same time increase erosion rates by sponges (Fang et al., 2013; Webb et al., 2017). Census approach has yet to include the role of several bioeroders (excavating sponges) (Murphy et al., 2016), which have been observed to become increasingly dominant on Caribbean reefs (Chaves-Fonnegra, Zea & Gómez, 2007). Moreover, ongoing ocean acidification appears to promote the contribution of chemical CaCO3 dissolution to total bioerosion by sponges even further (Duckworth & Peterson, 2013; Wisshak et al., 2013). Lastly, Silbiger & Donahue (2015) suggest that, under future climate conditions of increased pCO2 and ongoing warming, dissolution of existing reef carbonates is likely to be more affected than the growing of new reefs as such. Together this implies that there is an urgent need for in situ determination of NCC at the ecosystem level.

Direct approaches to accurately quantify NCC generally rely on determining the flux of alkalinity between water column and reef (Smith & Key, 1975). For reefs in environments characterized by a relatively linear flow of water over the reef, the upstream/downstream method (Odum & Hoskin, 1958; Gattuso et al., 1996) can be employed to determine NCC (Shaw et al., 2014; Koweek et al., 2015; Albright et al., 2016). For less unidirectional flow regimes, estimates based on overall residence time and knowledge of offshore conditions is needed (Courtney et al., 2016). In environments where low turbulence allows buildup of appreciable chemical vertical gradients, these gradients have been used to calculate net fluxes (McGillis et al., 2011; Takeshita et al., 2016). For fully exposed reefs, where no measurable accumulation may occur even in the boundary layer, the use of incubators is necessary. Several such incubation methods have been designed and applied. Most incubators cover a limited area (Patterson, Sebens & Olson, 1991; Haas et al., 2013; Camp et al., 2015), allowing single-species incubations. In most cases, numerous incubations are necessary to accurately capture variability between different locations on a reef in accretion/erosion and thus accurately estimate whole ecosystem NCC. When employing small-volume incubators, care must also be taken to maintain representative hydrodynamic conditions for the incubation species. Moreover, incubations must be terminated before NCC becomes depressed (for example by depletion of oxygen). Larger incubation structures (Yates & Halley, 2003) better capture variability on a community scale and convey environmental hydrodynamic conditions (surge) which, on the other hand, may cause inadvertent leakage of enclosed water. This potential exchange between ambient and enclosed water complicates the interpretation of observed chemical changes, particularly for signals that take relatively long to manifest themselves (e.g., alkalinity). The latter limitation restricts this method into hydrodynamically favorable (i.e., calm) conditions (McGillis et al., 2011). Additionally, due to obvious logistical challenges, most or all in situ incubations have been carried out on the reef flat.

Here, we aim to assess diurnal coral reef metabolic rates by investigating the in situ inorganic carbonate system over a reef slope coral reef patch offshore from the island of Saba, Dutch Caribbean. We use a tent-based incubation system in an environment with relatively strong currents and swell-induced near-bottom surge that caused modest exchange between the incubated reef and ambient water. Exchange between the enclosed and surrounding seawater is used to our advantage as this maintains oxygenated conditions during the incubation, thus allowing for increased incubation periods. Precise monitoring of temperature and salinity, both inside and outside the tent, allows for accurate determination of the amount of exchange across the enclosure. Explicitly accounting for the role of ambient variability, the benthic fluxes originating within the tent are inferred with high accuracy. Comprehensive monitoring of CO2 system parameters (dissolved inorganic carbon, total alkalinity), dissolved oxygen, and nutrients (phosphate, nitrate, nitrite, and ammonium) allows for subsequent quantification of integrated whole ecosystem coral reef metabolic processes (NCC and net community production) in a highly hydrodynamic environment.

Methods

Site and substrate

Reef incubations were performed west of the island of Saba in Ladder Bay at the Ladder Labyrinth mooring site of the Saba National Marine Park (17.6261°N, 63.2602°W) (Field permit approved by the Dutch Ministry of Infrastructure and Environment: RWS-2015/38370), between October 26th and 29th 2015, at a depth of 21 m (Fig. 1). The reef in this area has a distinct spur-and-groove morphology, and is located on a steep incline from the heights of Saba toward the ∼500 m deep stretch between the island and the Saba Bank carbonate platform. Coral reefs around Saba harbor a relatively rich diversity of marine species in the context of the wider Caribbean (Etnoyer, Wirshing & Sanchez, 2010). The location at which the tent was placed, was chosen such that a patch of coral reef with a community representative of the wider area was fully enclosed (Fig. 2). Within the 2 × 2 m enclosure, one larger and several smaller carbonate structures were present, acting as main substratum for benthic biota, together resulting in a in a total hard surface area ∼4.4 m2 and a 14% surface enlargement (rugosity). Abiotic components (sand and bare rock) accounted for 61% of the total surface area within the enclosure. Algae (algal turf, Lobophora spp., Dictyota spp.) covered 22%, sponges (among others Agelas sp. and Callyspongia plicifera) covered 7% and calcifying species such as corals (including Orbicella faveolata, Meandrina meandrites, and Diploria clivosa) and crustose coralline algae covered 4.2, and 6.6% of the total surface area, respectively (Fig. 2). No macro-bioeroding organisms were visible. A small number of heterotrophic animals, including small fish, crustaceans, and nudibranch, were present during the time of the incubation.

Figure 1 Area where the experimental enclosure was achieved with a schematic outline of the tent incubation.

(A) Photograph depicting tent, umbilical cord, support divers, and the dingy used for sample collection. (B) Schematic layout of the tent and surroundings. The Island of Saba is located to the east of this location. (C) Schematic side view of the employed setup for enclosure of a small patch of coral reef. The central structure is a rigid, inflatable dome tent, held securely in place by lead bricks and guy-lines (not shown). Inside the tent are located a battery powered mixing propeller for maintaining water circulation, and analyzers for salinity (S), temperature (T), oxygen (O2), and PAR. External to the enclosure are located another S/T analyzer, a current profiler and a pump (powered intermittently from the surface) which through an umbilical cord delivers enclosure interior water to the sea surface for sampling. Sampling of exterior water was performed either by this pump (with SCUBA divers temporarily disconnecting the connection to the tent interior) or by divers using large volume syringes. Zippers allow for opening of tent windows for re-equilibrating interior and exterior conditions between incubations.

Figure 2 Overview of the enclosed coral patch, after removal of tent at end of experiment.

Overview of the enclosed coral reef patch, after termination of the tent incubation. Yellow lines mark the extent of the tent (approximately 2 × 2 m).

Enclosure

The incubation enclosure is a custom-made, semi-hemispherical, bottomless, transparent dome tent with a square, four m2 footprint and ∼3.2 m3 volume. The tent walls consist of transparent polyvinylchloride of 0.8 mm thickness, with nontransparent reinforcements along the edges. The tent was inflated (on sandy sediment) by pumping water into the ribs of the dome, after which the rigid tent was carefully moved in place over the coral mound. Flaps extended ∼50 cm outward from each of the tent’s four sides, allowing for proper sealing of the tent to the substrate by placing weights on the flaps. All four sides of the tent contained an opening of ∼0.3 m2 to allow flushing of the enclosed volume between incubations: during each incubation this opening was sealed. Water enclosed in the incubation tent was homogenized by a continuously running propeller pump (model PP20; Jebao Ltd., Zhongshan, China). This pump was positioned close to one of the tent arches, at half the height of the tent, and generated a slight circulating turbulence, while minimizing stirring up of sediment. Effectiveness of the stirring was demonstrated by rapid and even dispersal of a small dose of injected fluorescein. Time required for initial deployment of the tent was approximately 4 h. In total, five incubations were carried out on this location, three during the day (incubations 1, 3, and 5) and two at night (incubations 2 and 4) (Table 1).

Table 1 Timing of the five incubations.

Incubation	1	2	3	4	5	
Latitude	17.6261°N	
Longitude	63.2602°W	
Depth (m)	21	
Start (local time)	October 27, 09:17	October 27, 17:10	October 28, 08:33	October 28, 17:15	October 29, 08:10	
End (local time)	October 27, 14:59	October 27, 22:13	October 28, 14:47	October 28, 23:20	October 29, 16:10	
Duration (min)	342	303	374	365	480	
Light mean (PAR)	65	3	58	3	91	
Std	23	1	14	1	15	
Minimum	22	2	11	2	5	
Maximum	120	8	75	8	161	
Note:

Incubation starting and end times, duration, and light. The listed PAR values are in μmol quanta m−2 s−1.

On the sandy substrate, adjacent to the main tent, a small secondary incubator was deployed. Its design is tetrahedron-shaped, and features transparent PVC-walled, rigid edges of one m, with 0.5 m long flaps extending from bottom edge. It covers a 0.43 m2 planar surface, and encloses a 118 l volume, resembling the cBIT described by Haas et al. (2013). Due to equipment constraints, only limited monitoring of this “sediment blank” incubator was performed by determining the total alkalinity (AT), total dissolved inorganic carbon (CT), and nutrient concentrations.

In situ measurements

Measurement of salinity (S), temperature (T), dissolved oxygen (O2), photosynthetically active radiation (PAR) and water current conditions within the large dome-shaped tent were performed throughout the duration of the incubations (4 h). S, T, and O2 were measured at 1 min interval using an actively pumped SBE37 MicroCAT equipped with an SBE63 optical dissolved oxygen sensor (Sea-Bird Scientific Inc., Bellevue, WA, USA). Drift of the involved sensors over the duration of our experiment was negligible, while precision (±1 × 10−5, ±1 × 10−4 °C, ±0.2 µmol kg−1, respectively) is orders of magnitude better than the changes in S, T, and O2 observed during incubations. PAR was assessed by an Odyssey light logger (Dataflow Systems PTY Ltd., Christchurch, New Zealand), calibrated in air against a superior instrument (Walz ULM500; Walz GmbH, Effeltrich, Germany). The MicroCAT and light logger were suspended from the apex of the enclosure at approximately half the tent’s height. A second CTD unit (model CastAway; YSI Inc., Yellow Springs, OH, USA) was deployed outside the tent to register ambient S and T during two out of the five incubations, due to logistical constraints.

Discrete sampling

During incubations, discrete samples were collected every 2 h for analysis of AT, CT, total organic carbon (TOC) and nutrients by pumping seawater from the tent interior (and, alternatingly, the exterior) up to the support vessel through a 50 m long 1/4″ Dekabon gas-impermeable “umbilical cord” (Fig. 1). The total volume pumped upward was ∼2 l per sampling event, after appropriate flushing (∼2 l) of the umbilical (internal volume ∼0.5 l).

Most analyses for AT were performed on-board (Caribbean Explorer II) using spectrophotometrically guided single-step acid titration (Liu et al., 2015). Additional samples for AT and CT were poisoned with HgCl2 immediately after collection (following Dickson, Sabine & Christian, 2007) for post-cruise analysis on a VINDTA 3C instrument (Mintrop et al., 2000). Accuracy of both instruments was set using certified reference material (batch 144) supplied by Scripps Institute of Oceanography (Dickson, Sabine & Christian, 2007). No appreciable bias in AT was apparent between the two instruments. On the VINDTA, a total of ∼125 samples were analyzed for CT and AT. Precision of replicates from the same sample bottle was 1.5 µmol kg−1 for CT and 1.0 µmol kg−1 for AT (for both instruments). However, precision for field replicates (i.e., replicates from separate bottles; n = 23) was 3.5 µmol kg−1 for CT and 5.0 µmol kg−1 for AT, possibly reflecting suboptimal sampling conditions and/or procedures (e.g., insufficient pre-flushing of umbilical before commencing filling of 1st replicate sample).

Samples for TOC determination were stored in pre-combusted 60 ml EPA vials and acidified and preserved with 8M HCl prior to shore-based analysis on a Shimadzu TOC-VCPN. Analytical precision for TOC (defined as standard deviation of differences between replicates) was ±9.9 µmol kg−1 (n = 8).

Samples for dissolved inorganic macronutrients (NO2+NO3, NO2, PO4, and NH4) were prepared by dispensing sampled water through 0.8/0.2 μm Acrodisc filters into five ml “pony vials,” and subsequently stored at −80 °C for later analysis at NIOZ on a QuAAtro continuous flow analyzer (SEAL Analytical GmbH, Norderstedt, Germany) following GO-SHIP protocol (Hydes et al., 2010). Uncertainty of nutrient determinations (±0.1, ±0.01, ±0.005, and ±0.005 µmol kg−1, respectively) was substantially smaller than the differences observed between samples taken over the incubation period.

Release of nutrients during respiration decreases AT (or increases AT for release of NH4+), confounding the interpretation of changes in AT to represent CaCO3 dissolution only. Following common protocol, we correct calculated AT for nutrient release as follows: ATobsNC=ATobs+PO4+NO3−NH4.

Throughout the remainder of the manuscript, AT equals ATobsNC as defined above.

Outline of data processing

After data collection, a six-step approach was taken to infer fluxes from the measurements. Numbered steps are discussed in more detail in the following sections. Briefly, (1) the leak rate of the enclosure is inferred from measurements of S and T performed simultaneously inside and outside the tent during two of the five incubations. (2) Assuming the inferred leak rate to be valid throughout the experiment (i.e., for the other three incubations as well), time series of exterior S and T are inferred for all incubations from tent interior S and T. (3) Time series of ambient concentrations of CT, AT, and O2 are predicted from linear relationships with salinity. (4) We calculate, accounting for leakage at a known and assumed constant rate, the time rate of substance input into the tent interior that best reproduce the observations made inside the enclosure. (5) We apportion the input of CT and AT into the contributions by the processes of CaCO3 dissolution and respiration. Lastly (6), all substance input rates are converted to fluxes.

(1) Rate of water exchange

The rate of water exchange across the enclosure f (in units of min−1) was estimated from the dampened response of measured in-tent salinity (S) to the variability of measured ambient (i.e., outside the tent) salinity over the duration of an incubation. This was performed by iterative minimization (based on least squares) of the residuals q in Eq. (1). (1) Sincalct+1+q=((1−f)⋅Sincalct+f⋅Sambientmeast)−Sinmeas

where, Sincalct + 1 is the calculated salinity inside the tent at time t + 1, Sincalct the calculated salinity inside the tent at time t and Sambientmeast the measured salinity outside the tent at time t.

(2) Ambient hydrography

With the estimated leak rate estimate f, an approximation of ambient salinity Sambientmeas may be obtained from Sinmeas, which is available for all five incubations.

(3) Ambient chemistry

In order to know, at high temporal resolution, the concentrations of O2, CT, and TA outside the enclosure, we regress measurements of these parameters against Sambientcalc. We use data collected (i) locally at the enclosure, supplemented by data obtained (ii) by vertical profiling down to ∼75 m in the vicinity of the incubator and (iii) during expedition PE414 of the Dutch RV Pelagia in Aug/Sep 2016 close to Saba (hydrographic station #49, <5 km from enclosure location; L. J. de Nooijer & S. M. A. C van Heuven, 2017, unpublished data). The use of data collected nearly a year later might be considered inappropriate. However, comparison between (i) Pelagia and (ii) tent ambient AT and CT data (and near tent profiles) is rather favorable.

(4) Time rate of substance input R

Having inferred (i) the (assumed constant) rate of exchange of water between tent and environment and (ii) the time history of ambient concentrations Cout of the parameter of interest (i.e., AT, CT, etc.) next we subsequently determined the constant time rate of substance input R (in µmol kg−1 h−1) that best explains the observed changes of concentration Cin inside the enclosure while accounting for constant exchange with the environment. This is performed through iterative minimization of the residuals q in Eq. (2).

(2) Cint+q=Cint−1⋅(1−f)+Cambientt−1⋅f+R

The inferred input rate R is somewhat sensitive to the choice of the initial interior concentration (Cint = 0). Using the measurement collected at the start of the incubation may affect the result due to stochastic measurement error. Therefore we used an initial Cin through careful observation of initial measurements performed in the enclosure and of the measured (and predicted) ambient conditions. The dictated initial interior concentrations were identical for all five incubations, supported by the observation of comparable ambient salinity at the start of each incubation.

An estimate of the robustness of the input rates of O2, CT, and AT is obtained using a Monte Carlo approach (Fig. S1). A thousand curve fits were performed as above, but after randomly perturbing (i) each of the measured values of CT and AT (both by samples from a normal distribution of widths’ of four µmol kg−1, representing the measurement precision), (ii) the times of collection of the samples (s = 5 min) and the leak rate of the tent (s = 0.1% min−1). If the standard deviation of the 1,000 obtained input rates was smaller than the calculated nominal input rate, this nominal rate is considered to be significantly different from zero.

(5) CaCO3 dissolution and respiration rates

As outlined above, measurements of AT haven been adjusted for the effect of nutrient release by respiration. Subsequently, the individual contributions of CaCO3 dissolution and respiration to the observed concentrations (or fluxes) of AT and CT were calculated: ΔATdiss = ΔATobsNC       change in AT due to dissolution

ΔATresp = 0         change in AT due to respiration

ΔCTdiss = ΔATobsNC/2       change in CT due to dissolution

ΔCTresp = ΔCTobs − ΔCTdiss)    change in CT due to respiration

(6) Conversion to fluxes

The input rates R (again, in µmol kg−1 h−1) in the tent are converted to fluxes (µmol m−2 h−1), assuming an enclosed mass of water of 3,000 ± 150 kg (approximately 3,200 l enclosed; substrate volume is ∼250 l; seawater density ∼1,022 kg m−3) and an incubated planar surface of 4.4 m2.

Lastly, we compare our results with NCC estimates based on observed community composition and data published for the fluxes of various classes and species of reef organisms, following the ReefBudget approach of Perry, Spencer & Kench (2008). To measure the species-specific cover within the incubated area, we took multiple photos from different angles and then used ImageJ v1.51j8 to quantify the precise cover of each functional benthic group. Rugosity was measured from four crossed transects through the incubated patch (see Tables S1–S3).

Results

Application of Eq. (1) to data collected during incubations four and five yields a leak rate of the enclosure f of ∼0.007 min−1. This indicates that ∼25 kg of seawater (i.e., 0.007 × 3,000 kg) is exchanged every minute between the incubation enclosure and the environment. Although f ∼0.007 accurately relates interior and ambient salinity observations made during incubations 4 and 5, we have no direct means of ascertaining that it also applies to preceding incubations. However, given the sparse ambient salinity data, we assume this water exchange rate to be constant throughout all incubations. Because the sealing of the tent to the substrate remained unchanged from the second incubation onward, the time history of ambient salinity Soutcalc was derived under the assumption of f being ∼0.007. During incubations the error between Soutmeas and Soutcalc was −0.023 ± 0.19 (range: 34.7–36.0). The good match indicates validity of this simple adjective exchange model.

The ambient hydrographic conditions reflect variable admixture of a deeper, colder, and more saline component into the warmer, and fresher waterbody that is more commonly encountered at the incubation site (see Fig. 3A). In these ambient waters, CT and AT increase with S as is expected. Oxygen, too, is observed to increase with increasing S, due to the higher solubility in the more saline and—crucially—colder water. Regressions between S and O2/CT/AT are presented in Fig. 3, panels B, D, and F. The time histories of these properties, derived using Sambientcalc, are presented in Fig. 3, panels C, E, and G. For nutrients, no regressions were performed since ambient concentrations were essentially invariant at zero compared to the in-tent changes during incubations (Fig. S2).

Figure 3 Measured conditions within and outside the tent.

(A) Salinity and temperature recorded outside the enclosure show occasional mixing of a cool, saline deep water body into the warmer, fresher surface water component that is most commonly observed at the depth of the enclosure (as indicated by the high density of data points at S∼34.8). (B, D, F) Regression against salinity of, respectively, AT, CT, and O2. Samples in these regressions originate from three sources (two for O2). (C, E, G) Measured values of, respectively AT, CT, and O2, plotted together with time trace of these values, generated from the regressions against salinity.

Figure 4 illustrates the results of our approach for incubation 4. For an overview of all results from all incubations, please refer to Table S4 and/or Fig. S3. The model employed fits the measurements for CT and AT relatively well: the RMSE of the fit of CT is 3.5 µmol kg−1—identical to the measurement uncertainty of CT itself. For AT the fit (5.3 µmol/kg) is slightly worse than instrument precision (3.5 µmol kg−1).

Figure 4 Results of the model employed to infer the input rates from observations.

Illustrative results of the model (bold red line) employed to infer the substance input rates from observations, here for incubation 4. The thin blue line depicts the predicted ambient values while the blue crosses represents the measured ambient values. (A) A water exchange rate of 0.007 min−1 between enclosure and environment optimally relates ambient and interior measurements of salinity. Back-calculating ambient salinity from interior salinity is shown here to be feasible. (B) Ambient temperature co-varies with S. From T, the exchange rate is inferred to be 0.012 min−1 (higher than from S due to additional conductive equilibration). Additionally, in (C–F), we present results of a simplistic asymptotic curve fitting (thin red line) to show how the two methods may diverge appreciably, here mostly evidently for AT. (Note that the uncertainty of the asymptotic fit is worse than that of the model used in this study).

For the five incubations, consumption of oxygen from the incubated seawater ranged from −10 to −30 µmol kgSW−1 h−1 (Table S4). Concomitant increase of CT, PO4, and NO2+3 strongly suggests respiration to be the dominant process throughout these incubations. Respiration should decrease AT slightly due to the release of nutrients, but an even larger decrease is inferred by the full model suggesting a significant role for net calcification during incubation 2, 3, and 4. For the incubation 1 and 5 on the other hand, slight CaCO3 dissolution is inferred. Prior to inferring rates, values of AT were adjusted to account for the release of nutrients during respiration. Therefore, by definition the respiration AT rate is zero for all incubations (Table 2).

Table 2 Summary of fluxes inferred.

Incubation	1	2	3	4	5	
PO4	0.015 ± 0.004	0.009 ± 0.005	0.016 ± 0.005	0.031 ± 0.009	0.037 ± 0.007	
NO2+NO3	0.49 ± 0.14	0.55 ± 0.21	0.83 ± 0.09	0.91 ± 0.18	0.81 ± 0.20	
NH4	0.21 ± 1.06	0.26 ± 0.48	0.18 ± 0.05	0.23 ± 0.04	0.17 ± 0.19	
NO2	0.02 ± 0.01	0.01 ± 0.01	0.02 ± 0.00	0.02 ± 0.00	0.01 ± 0.01	
SIL	0.44 ± 0.10	0.00 ± 0.18	0.19 ± 0.10	−0.07 ± 0.20	−0.02 ± 0.15	
DOC	1.4 ± 4.3	−1.1 ± 4.2	−0.9 ± 4.8	−1.9 ± 2.8	−0.1 ± 5.4	
O2	−9.0 ± 0.7	−13.5 ± 1.1	−13.6 ± 1.3	−21.7 ± 1.9	−17.0 ± 1.6	
CT	5.0 ± 0.8	11.1 ± 1.1	13.0 ± 1.0	17.1 ± 1.6	10.6 ± 1.1	
AT	0.4 ± 0.6	−2.5 ± 0.9	−2.8 ± 0.6	−0.4 ± 0.7	0.4 ± 0.5	
CTresp	4.8 ± 0.8	12.3 ± 1.1	14.4 ± 1.0	17.3 ± 1.6	10.4 ± 1.1	
ATresp	0.0 ± 0.0	0.0 ± 0.0	0.0 ± 0.0	0.0 ± 0.0	0.0 ± 0.0	
CTdiss	0.2 ± 0.3	−1.2 ± 0.4	−1.4 ± 0.3	−0.2 ± 0.4	0.2 ± 0.2	
ATdiss	0.4 ± 0.6	−2.5 ± 0.9	−2.8 ± 0.6	−0.4 ± 0.7	0.4 ± 0.5	
Notes:

Summary of fluxes inferred from observed concentration changes in the enclosure during each of five incubation periods. See text for methods for derivation of rates and uncertainties. All fluxes in mmol h−1 m−2 (planar incubated surface area). Negative dissolution fluxes are described (instead of positive calcification) to maintain co-directionality of the fluxes of respiration and dissolution.

Uncertainties for respiration and dissolution are assumed to be equal to uncertainties in CT and AT, respectively.

Results obtained from incubations using the smaller, secondary tent suggests at most a very limited role for sedimentary processes (see Fig. S4): although exterior concentrations of AT and CT increased by as much as 60 µmol kg−1 in the second half of the incubation, interior AT and CT during those 4 h did not increase at rates higher than 2.5 µmol kg−1 h−1. This limited response suggests a leak rate between 0.2% and 1% min−1. Appreciable accumulation may therefore be expected if fluxes are present, also because of the small tent’s favorable volume-to-surface ratio (275 vs 750 l m−2 for the dome tent). Over the first 165 min of the incubation, however, interior AT and CT increase only by 1 and 2 µmol kg−1 compared to initial conditions (commensurate flux: ∼0.3 and 0.6 mmol m−2 h−1, that is, one or two orders of magnitude lower than in the main tent; note that appreciable errors apply both to measurements and conversion to flux). These increases may be linked to exchange with ambient waters. No changes in pyramid tent nutrient concentrations were observed, suggesting low rates of productivity or respiratory processes. For subsequent calculations, we consider the contribution by sedimentary processes to be negligible.

Subsequently, the concentration changes in oxygen, inorganic carbon, etc. in the dome-tent were used to calculate the fluxes during each of the five main incubations (Table 2). No trends were observed in TOC concentrations within the tent during incubations, despite clear trends being observed for CT.

Deviations of the tent leak rate from the nominal 0.7 % min−1 (or “breach events”) are observed during all incubations. These deviations from our assumption may negatively affect the inferred rates of change in O2 concentrations, as evident from the differing slopes of trace and fit near the starts of incubations. Considering only the first 30 min of the oxygen traces (preceding tentative breach events in all incubations), appreciably higher oxygen consumption is inferred than when considering the full traces (Table S5). At constant ambient S, O2, CT, and AT, leakage at a rate higher than the 0.7% min−1 that we assume would result in underestimation of the true fluxes of O2, CT, and AT. No change would be observed in the O-flux/C-flux ratio that we infer. Conversely, at times of sudden high ambient S, O2, CT, and AT, leakage at a rate higher than the 0.7% min−1 that we assume would result in overestimation of CT and AT fluxes, and underestimation of O2 fluxes. This would change the O-flux/C-flux ratio that we infer. However, as we do not know if the O2 trace is asymptotic, or that the tent did indeed leak at higher or lower rates than 0.7% min−1, results are likely still valid (i.e., when no breach events occurred, and O2 deviations resulted from sensor artefacts or true biological activity). We therefore maintained the conceivably affected incubations in the paper.

The NCC inferred from our incubations ranges from −0.2 to 1.2 kg CaCO3 m−2 year−1 which is on average higher but still in range than the NCC estimated from the ReefBudget method (0.36 kg CaCO3 m−2 year−1, Table 3).

Table 3 Net community calcification estimates from flux-based method and ReefBudget method.

Incubation	1	2	3	4	5	
NCC(this study) (kg CaCO3 m−2 year−1)	−0.2 ± 0.3	+1.1 ± 0.4	+1.2 ± 0.3	+0.2 ± 0.4	−0.2 ± 0.2	
NCCReefBudget (kg CaCO3 m−2 year−1)	0.36	0.36	0.36	0.36	0.36	
NCC(this study) (mmol CaCO3 m−2 h−1)	−0.2 ± 0.3	+1.2 ± 0.4	+1.4 ± 0.3	+0.2 ± 0.4	−0.2 ± 0.2	
NCCReefBudget (mmol CaCO3 m−2 h−1)	0.41	0.41	0.41	0.41	0.41	
Note:

NCC of the planar total area inside the incubation calculated from fluxes and the ReefBuget method.

Discussion

Our results indicate that the tent incubation is an effective tool for in situ quantification of reef fluxes in reef-overlaying water. Quantification of fluxes was achieved despite strong variability in ambient conditions and in the presence of appreciable swell-induced seawater exchange. To this end we applied a comprehensive conceptual framework for the interpretation of the measured concentration differences. This method allows for a volume exchange between the environment and the incubation thereby replenishing the latter and keeping the O2 levels within the tent near ambient conditions resulting in minimized unrepresentative reef community metabolism. By continuously monitoring the inside environment and assuming constant exchange rate, fluxes within our incubation can be treated as if acquired by a flow through system. Nonetheless, future application of this or similar incubation methods could be further improved by continuous monitoring of the exchange rate, rather than assuming it to be constant throughout the incubation. This could be obtained for instance by running a second thermosalinograph outside the tent. The application of a conceptually simpler “asymptotic” model yields different and less well-constrained results. Particularly, the direction of the CaCO3 dissolution flux may be seen to be reversed in the simpler method (see also Fig. 4). In all incubations, for both AT and CT, the uncertainty in measured concentration differences and the variability between results may be greater for the simpler model (Table S4; Fig. S3). In the case of AT (Fig. 4F), the assumed-to-be-constant input of AT inferred by the model applied here is of opposite sign to the simpler asymptotic model result (−0.6 vs +7.1 µmol kg−1 h−1). This reversal of sign of the AT rates observed between the two models (asymptotic and full) is caused by the inability of the asymptotic model to account for occasional intrusion of high-AT ambient water into the tent during sudden changes in ambient hydrography. The asymptotic model (panels C and D) shows a relatively good fit of the observations of O2 and NO2+3 around the fitted curve, which is due to the invariant ambient concentrations of these parameters.

In contrast with previous studies carried out at shallower depths using either Lagrangian drifts or incubations, all net diurnal rates from this study are strongly skewed toward respiration suggesting net heterotrophy in all incubations. Studies performed at shallower depths shift between net autotroph and net heterotrophy over the course of a day (Yates & Halley, 2003; Albright, Langdon & Anthony, 2013; Albright et al., 2015). However, previously reported average net respiration rates occurring at night on shallower reefs are comparable to results from this study (14.5–35.5 mmol C m−2 h−1). Results are also comparable to previously reported values at depth. For example, Middelburg, Duarte & Gattuso (2005) compiled a global mean coral reef respiration rate of 131 ± 46 mol C m−2 year−1. Specifically for their categories “outer reef slopes” and “high activity areas” they report values of 140 ± 70 and 413 ± 187 mol C m−2 year−1, respectively. This range compares well to the rates reported here (105–298 mol C m−2 year−1 for the full planar surface). A stoichiometric comparison of the inferred fluxes of CT, oxygen and nutrients is combined with (i) the canonical “Redfield ratio” (Redfield, 1963) of the elemental composition of open ocean phytoplankton and (ii) the median elemental composition of benthic macroalga (Atkinson & Smith, 1983), likely resembling the composition of the labile fraction of the locally present organic carbon (Table 4). This shows that the community incubated in this experiment respires carbon and nutrients in a ratio that resembles the composition of benthic macroalgae (Atkinson & Smith, 1983) which indicates that the observed signal is indeed originating from the sedimentary, benthic, macrofaunal, and/or bacterial constituents of the enclosed community.

Table 4 Stoichiometry of rates observed during five incubations.

Incubation	1	2	3	4	5	All	R1963	A&S1983	
P	1	1	1	1	1	1	1	1	
N	32.8 ± 12.2	61.2 ± 42.1	51.5 ± 17.9	29.7 ± 10.7	21.7 ± 6.5	33.2 ± 5.6	16	30	
O	337.5 ± 99.7	1222.4 ± 714.9	805.8 ± 275.4	557.9 ± 175.1	313.3 ± 60.8	535.6 ± 73.3	106	550	
C	−608.7 ± 157.5	−1485.3 ± 867.5	−842.9 ± 289.4	−706.6 ± 223.6	−455.2 ± 91.6	−691.7 ± 94.9	−150	−610	
O/C	−1.80 ± 0.33	−1.22 ± 0.14	−1.05 ± 0.13	−1.27 ± 0.14	−1.45 ± 0.18	−1.29 ± 0.07	1.22	NA	
Note:

Stoichiometry of rates observed during five incubations. Values in column “avg.” are calculated as the ratio of the sums of incubations 2–5. Uncertainties are calculated by error propagation. Rightmost two columns show literature values from (i) Redfield (1963) and (ii) Atkinson & Smith (1983), representing the elemental compositions of (i) marine phytoplankton and (ii) benthic macroalgae. Our incubation results most closely resemble the latter.

A strong correlation between NCC and net community productivity in reef environments is well documented (Gattuso et al., 1996; Shaw, McNeil & Tilbrook, 2012; Shaw et al., 2015; McMahon et al., 2013; Albright et al., 2015), however, no correlation was found in our incubations. The NCC inferred from our incubations ranged from −0.2 to 1.2 kg CaCO3 m−2 year−1 which is on average higher than the mean recorded rate of 0.2 kg CaCO3 m−2 year−1 associated with a Floridian reef patch (10% coral cover) in shallower waters (Yates & Halley, 2003) using a similar method.

Applying the ReefBudget approach to our benthic census data for comparison, we obtain a NCC for the full incubated substrate surface (i.e., sand and hard substrate) of ∼0.36 kg CaCO3 m−2 year−1 (see Table 3). This estimate is slightly lower but comparable to the average NCC inferred from our incubations (∼0.42 kg CaCO3 m−2 year−1, Table 3). The range of NCC estimates inferred from our results indicates how sensitive metabolic and chemical processes on coral reefs are to their environment. The chemical flux-based method as presented here is appreciably sensitive to the effects of the surrounding hydrological conditions on the substrate and this may be the source for a slight discrepancy compared to the ReefBudget approach. NCC rates acquired by Perry et al. (2013) associated with a coral cover ranging between 4% and 5% and at similar depth (17–20 m) in the Bahamas are all negative (ranging from −0.01 kg CaCO3 m−2 year−1 to −0.23 kg CaCO3 m−2 year−1). This may be explained by varying community composition such as the absence of macro-bioeroders within our tent. Furthermore, the flux-based method does not assess the mechanical component of bioerosion (caused by parrot fish or sponges for instance) which is important to the process of reef accretion.

The ReefBudget method offers a fast and convenient tool for estimating reef biogenic carbonate production states both on a remarkable temporal and spatial scale. Although the incubated flux-based approach, may be more sensitive to unstable and varying reef states, it cannot offer such a large spectrum of study. However, it provides an assessment of the full community without having to determine calcification/dissolution rates as a function of surface area. This can be very useful to assess the effect of endolithic species or determine the impact that some understudied organism may have on the chemical conditions. For instance, benthic cyanobacterial mats have been shown to proliferate around the islands of Curacao and Bonaire since 2003 (De Bakker et al., 2017) and are described to effect pH on a local scale (Hallock, 2005; Paerl & Paul, 2012). However, close to no records on how these mats may alter reef chemical conditions and subsequently impact the calcifying/bioeroding community are available. Currently, the ReefBudget approach relies on various assumptions regarding the calculation of each biological component. As such, the flux-based approach described here should not be regarded as a substitute for survey methods such as the ReefBudget, but rather as a complementary tool. Using the flux-based approach, it will become easier to determine missing components and variations in chemical dissolution/calcification on a spatial (e.g., depth) and also smaller temporal scale (i.e., diurnal cycle, seasonality), therewith improving survey based carbonate budget assessments.

To determine if the respiration signal might be an artefact of the incubation treatment, we identify potential causes that may perturb the signal. The estimated contribution by macrofauna such as fish, crustaceans, and nudibranchs to the observed respiration signal is deemed to be negligible: considering a fish mass-specific O2 consumption rate of ∼100 mgO2 h−1 kg−1 (Roche et al., 2013), and assuming 100 grams of fish to be present in the tent (which is likely a strong overestimate as only very few small fish were observed during incubations), we calculate a contribution to CT in the incubation of ∼0.1 µmol kg−1 h−1, which is two orders of magnitude smaller than the observed respiration rates of 10–30 µmol kg−1. Additionally, we rule out that “free floating” TOC (e.g., coral exudates) is the material that is respired. While clear fluxes are inferred for CT, no trends were observed in TOC concentrations within the tent during incubations. Alternatively, no significant depression was observed of average interior TOC values (84 ± 9 µmol kg−1, n = 39) relative to exterior TOC (86 ± 7 µmol kg−1, n = 8). Although the tentative drop in TOC resembles the small drop observed in dedicated dissolved organic carbon (DOC) depletion experiments (De Goeij & Van Duyl, 2007), the lack of volume flow through the tent means TOC cannot be more than a very minor source of respirable carbon. Absence of depletion of suspended labile TOC notwithstanding, TOC may still play a role in the form of mucus if that is adhered to substrate or incubator, out of reach of sampling but available for bacterial respiration. However, the respiring biomass required for the observed CT increases is unlikely to be present in the form of bacteria, especially shortly after incubation start. Indeed, Wild et al. (2004) show from small-scale incubations that the bacterial degradation of coral mucus, introduced into their incubators (containing only sediment and water column) at high concentrations, occurs at rates of 0.7–2.1 mmol m−2 h−1. That compares to rates around 60–175 mmol m−2 h−1 observed in our experiment, suggesting remineralization of adhered mucus plays at best a minor role in our incubations, further suggesting the observed fluxes to originate from the macroscopic biotic substrate.

In that category, sponges are the most likely organisms respiring, having appreciable biomass and containing ample energy stores to maintain respiration during the incubation periods, in which only limited amounts of organic carbon are available for filter feeding. Hadas, Ilan & Shpigel (2008) report (Red Sea) sponge basal oxygen consumption to be ∼50% of consumption featured during full water pumping activity, which means that sponge respiration largely continues even when filter feeding ceases. These authors report a rate of 2.4 µmol O2 h−1 g−1 (wet weight). Similarly, Ludeman, Reidenbach & Leys (2017) report sponge oxygen consumption (standardized to sponge volume) ranging from 0.3 to 3 µmol h−1 ml−1, with strong species dependence. Assuming the higher end of this range applies to the sponges incubated in our experiment (mostly Agelas sp., C. plicifera), and assuming as much as five kg wet weight of sponge to have been present in the enclosure, we account for ∼5 µmol kg−1 h−1 of the observed rates of ∼30 µmol kg−1 h−1. Maintained respiration by sponges throughout the series of incubations could be fueled by filter feeding during the ∼50% of the time in which the incubator was open to the ambient water. Recent research by McMurray et al. (2018) showed that species hosting abundant symbiotic microbes (i.e., high microbial abundance, HMA) primarily consumed DOC, while the diet of species with low microbial abundances (LMA) primarily consisted of detritus and picoplankton. They further pointed out that it remained unknown if DOC released by LMA species could be a source of food for HMA species. The main sponges incubated in our experiment are Agelas sp. and C. plicifera and represent respectively a HMA and a LMA sponge. We tentatively infer that this may explain partly the observed high respiration rate. Nevertheless, we cannot infer if sponges are able to maintain metabolic balance throughout the incubation period, or that they deplete their stores. Further analyses such as bacterial counts would be needed to answer such questions. From Table 2 we conclude oxygen consumption rate during daytime (incubations 2 and 4) to be lower by ∼5 µmol kg−1 h−1 than night time rates (incubations 3 and 5), hinting at a role of primary producers (corals, CCA, macro and microalgae).

The lack of accumulation of CT in the secondary, small incubation places our sediment at the very low end of literature values regarding respiration. For example, Middelburg, Duarte & Gattuso (2005) report a global mean sediment respiration value of ∼8.5 ± 7 mmol C m−2 h−1 (as approximated from their Figure 11.3). Our observed low values may be reasonable considering the highly hydrodynamic nature of the incubation environment which likely hampers settlement of substantial amounts of organic matter onto and into the sediment. In addition, the volcanic sandy composition of the sediment around Saba may be less prone to dissolution than coralline sediment and could explain the insignificant increase in AT in the small tent. However, Eyre et al. (2018) shows similar results for sediment around Cook Islands which is mostly composed of calcareous fragments (Wood, 1967). Eyre et al. (2018) shows that dissolution in reef sediment across different locations around the world is negatively correlated with the aragonite saturation state (Ωar). Average Ωar of ambient water around the tent incubation throughout the experiment is calculated to be 3.85 which is more comparable to islands (Bermuda and Tetiaroa) showing accretion in reef sediment. The combined effects of hydrodynamics and sand composition are likely to explain why our results present neither accretion nor dissolution in our tent’s sediment.

Conclusions

Flux-based carbonate budget studies, as presented here, provide quantitative data on the functional state of reefs in terms of biologically driven carbonate production which is particularly sensitive to ambient environmental conditions. As such, they can be particularly useful for temporal studies, especially to reveal not only diurnal and seasonal patterns but also to capture shifts in functionality of reef systems. We incubated a coral reef patch situated in a high-energy environment which caused a limited amount of seawater exchange. Monitoring of conditions within and outside the tent allowed for determination of the exchange rate and thereby allowed for correcting the respiration and calcification rates. Application of this procedure shows that this reef patch is characterized by NCC inside the tent at a rate within range but on average higher than fluxes reported in previous studies for shallower reef systems indicating coherence in our results. However, the range of NCC estimates inferred from our results accounts for the sensitivity of this reef patch to the surrounding environment. Furthermore, the net heterotrophy reported here both during the day and the night differs from studies performed at shallower depths where shifts between net autotrophy and net heterotrophy are observed. Future research may include various types of substrates and comparison between regions with varying water quality.

Supplemental Information

Supplemental Information 1 Total calcification using the ReefBudget method.

ReefBudget method applied to the 4.4 m2 full planar surface of the tent. [Calcification Rate] = [Cover] * [Specific Calcification Rate] * [Rugosity] and NCCReefBudget = [Total Coral Calcification] + [Total CCA Calcification]–[Total Microbioerosion].

Click here for additional data file.

Supplemental Information 2 Surface area of substrate classes.

Click here for additional data file.

Supplemental Information 3 Rugosity, planar area and 3D area.

Rugosity. Planar areas were determined from photogrammetry, based on photographs collected after incubator removal at end of the experiment. Rugosity of 1.36 was determined in the field after removal. Rugosity of the sandy part of the patch was assumed to be approximately 1.00. Average rugosity of the patch is calculated.

Click here for additional data file.

Supplemental Information 4 Calculated rates within the enclosure.

Rates in μmol kgSW−1 hour−1; the s.e. of the fit; and the rmses of the fits.

Click here for additional data file.

Supplemental Information 5 O2 change rates.

Comparison of rates of change in O2 concentrations inferred from either the full O2 concentration history (third column) or from only the first 30 minutes of each incubation (rightmost column). The latter method provides a more consistent separation between day- and night time incubations (especially when ignoring incubation 1). The daytime average O2 consumption is ∼20% below the night time rate.

Click here for additional data file.

Supplemental Information 6 Method to assess robustness of the inferred rates.

Example of results of the ‘Monte Carlo’ method used to assess robustness of inferred time rates of change of concentrations during incubation. Here, measured values of AT and the leak rate are varied slightly (±4 μmol kg−1 for AT, ±0.1 % min−1 for leak rate), and curves are repeatedly fit. The average and standard deviation of one thousand such fits are presented. In this example, the rate average is slightly larger than the associated uncertainty, and the rate is thus assumed to be significantly different from zero.

Click here for additional data file.

Supplemental Information 7 In-tent and ambient nutrients.

Interior and ambient nutrient concentrations.

Click here for additional data file.

Supplemental Information 8 Data from secondary ‘pyramid’ incubator, placed on bare sediment.

In-tent measurements and model fits of all five incubation periods. Second and fourth incubations were during nighttime. The legend presented in the third row of panels applies to all subsequent rows (except PAR, which was measured).

Click here for additional data file.

Supplemental Information 9 Example of results of the ‘Monte Carlo’ method used to assess robustness of inferred time rates of change of concentrations during incubation.

Data from secondary ‘pyramid’ incubator, placed on bare sediment. The observed trends are the result of the (unknown) balance between (i) exchange with the environment and (ii) sedimentary processes. Irrespective of dominant process, sedimentary fluxes are inferred to be negligibly low compared to those observed in the primary incubator (see main text).

Click here for additional data file.

The authors are particularly grateful to the captain and crew of Caribbean Explorer II, who have been outstandingly helpful and accommodating. We also would like to thank the institutional support of Saba Marine Parks, Caribbean Netherlands Science Institute (CNSI), and Wageningen Marine Research (WMR). Additional gratitude is reserved for Janine Nauw, Johan Stapel, Steve Piontek, and volunteer divers Oscar Bos, Dahlia Hassell, Jarno Knijff, Ewan Tregarot, Lodewijk van Walraven, and Bas Westerhof.

Additional Information and Declarations

Competing Interests

Author Contributions

Field Study Permissions

Data Availability

The authors declare that they have no competing interests.

Steven M.A.C. van Heuven conceived and designed the experiments, performed the experiments, analyzed the data, contributed reagents/materials/analysis tools, prepared figures and/or tables, authored or reviewed drafts of the paper, approved the final draft.

Alice E. Webb conceived and designed the experiments, performed the experiments, analyzed the data, contributed reagents/materials/analysis tools, prepared figures and/or tables, authored or reviewed drafts of the paper, approved the final draft.

Didier M. de Bakker performed the experiments, analyzed the data, contributed reagents/materials/analysis tools, prepared figures and/or tables, authored or reviewed drafts of the paper, approved the final draft.

Erik Meesters performed the experiments, analyzed the data, contributed reagents/materials/analysis tools, authored or reviewed drafts of the paper, approved the final draft.

Fleur C. van Duyl conceived and designed the experiments, performed the experiments, analyzed the data, contributed reagents/materials/analysis tools, prepared figures and/or tables, authored or reviewed drafts of the paper, approved the final draft.

Gert-Jan Reichart analyzed the data, prepared figures and/or tables, authored or reviewed drafts of the paper, approved the final draft.

Lennart J. de Nooijer conceived and designed the experiments, analyzed the data, prepared figures and/or tables, authored or reviewed drafts of the paper, approved the final draft.

The following information was supplied relating to field study approvals (i.e., approving body and any reference numbers):

Field experiments were approved by the Dutch Ministry of Infrastructure and Environment (registration number RWS-2015/38370).

The following information was supplied regarding data availability:

4TU.Center for Research Data: https://data.4tu.nl/repository/uuid:d7c6503d-cd91-4a8d-8d50-8583dd215a14.

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
