# Peer review of "In-situ incubation of a coral patch for community-scale assessment of metabolic and chemical processes on a reef slope"

_PeerJ, doi:10.7717/peerj.5966_

## Round 0.1 · original submission · Major Revisions

Your manuscript has now been reviewed by two noted experts in the field of coral reef carbonate dynamics, both of whom provided thorough and excellent reviews that will clearly improve your work. Both commended your efforts to work on reef slopes in measuring metabolism of reefs, and it is clear that this aspect of the work should be brought to the forefront; I would ask you to consider revising the title and abstract to reflect the nature of your findings rather than the design of your methods (this is often called an "active title"). Your abstract and introduction should emphasize the hypotheses being tested, and Reviewer 2 noted that the observations of net heterotrophy should be emphasized as interesting and novel. However, such observations depend on accurately assessing how the tents "leak" and the comments by Reviewer 1 on this aspect will be critical to address in assessing if this heterotrophy is real.

Reviewer 1 in particular expressed considerable frustration with the clarity of presentation of your benthic fluxes. There is no reason for this inherent to your study design, and I recommend firmly that you carefully address these recommendations to provide rates of net community calcification and production with all of the relevant calculations transparent. Visualization of the data is central to communicating your points, and I would guess that a revised manuscript would contain several efforts to revise figures and/or add more figures to better distill the dynamics inside and outside of the tents to clarify these important details for the reader.

I would expect that the revision carefully consider and address each of the reviewers' excellent points - we are very lucky to have their specific expertise on this manuscript and their points are all strong. In the rare case that you disagree with a reviewer opinion I would expect a clear justification of why you have opted against the recommended revision. I would expect that the revised manuscript will be subject to additional review, and there is no guarantee of eventual acceptance.

We look forward to your revised manuscript.

Reviewer 1 ·

Basic reporting

In this study, the authors placed a small, isolated patch reef and a section of bare sediment in large and small benthic incubation tents respectively to measure day and night benthic metabolic rates near the island of Saba. The authors incorporated small amounts of seawater exchange across the tent boundary throughout the experiment into mixing models in order to quantify reef metabolic rates within each tent. Making in situ metabolic measurements at depth, and especially at night!, is a difficult task and I commend the authors for the great amount of work involved in this study. This type of work is incredibly valuable for our understanding of coral reef metabolic rates and is critical for pushing scientific understanding of coral reef processes forward. However, I have a few concerns with this manuscript especially with respect to the leakage of the tents and method descriptions in this study that should be addressed prior to publication.

(1) The study introduction discusses investigating the carbonate chemistry of an offshore patch reef with much attention to NCC and NCP but does not include clearly testable hypotheses. Please clearly state the hypotheses being tested in this manuscript in the introduction and discuss whether the evidence supports or does not support the hypotheses tested in the discussion.

Experimental design

(1) My biggest concern with this manuscript is the leaking of the tents and how they were accounted for in the models as much of the text appears to contradict itself with respect to how well constrained these leaks were in the models. For example, line 242 implies that the mixing model was derived for only 20% of the time of the incubations. How do you know mixing rates for the time that measurements were not conducted both inside and outside of the chamber? The authors later state in line 395 that there are occasional large breach events that have not been accounted for in the model for O2? How does this not also affect the conclusions of the other parameter fluxes? In line 316, the authors state the O2 is an excellent marker for the seal of the tent but then state that only incubations 3-5 provide O2 data that allows asymptotic leak rates to be assumed? Is this for the asymptotic model only? If the assumptions for the asymptotic model are invalidated across 2 of the 5 incubations, then why is this model employed and described in the manuscript? Line 430 then states that depending on the method to account for leakage used, either net calcification or net CaCO3 dissolution was measured, which further makes me question whether there’s a strong understanding of what actually happened in the tents based on the modeling methods employed in this study. As stated in the section 1 of this review, some of this confusion could in part be due to the language used throughout the manuscript in addition to alternating between discussions of the large patch reef tent and the small sediment tent and discussions of two mixing models employed in this study. Nonetheless, the manuscript appears to contradict itself in these statements regarding the most central methodology of this manuscript. The authors recommend these tents as critical tools to be used elsewhere for metabolic measurements, but the contradicting statements in the manuscript suggest there may be critical flaws in the methods and their interpretation. These points and apparent contradictions need to be addressed before publication.

(2) I do not follow the logic in line 437. Does the “5-25 kg CaCO3 m-2 yr-1 calcifier surface” refer to calcification rates of individual coral taxa within the tent? As the paper is written, it’s not clear how the ReefBudget portion of the study was carried out and how to compare these rates to the chamber flux measurements in this study. Please include a table summarizing the %-cover by each taxa, the rates of CaCO3 production, and resulting taxa-level contribution to the ReefBudget numbers to show how these ReefBudget numbers were derived and include a more formal comparison between this ReefBudget calculation to the NCC in this study before publication.

(3) In Table 4, how were the dissolution fluxes attributed to the respective surfaces within the 4 m2 patch? Additionally, why are negative dissolution fluxes described instead of stating that positive calcification was observed? The authors state that rates for the incubations were 0.63±0.46 kg CaCO3 m-2 yr-1, but I do not see the respective NCC rate for each incubation, nor how it was normalized to biotic surface areas. The authors allude to some aspect of ReefBudget methodology but this is not clear. Please describe the benthic flux attributions, ReefBudget methodology, and more clearly state the NCC rates for each incubation before publication.

(4) In line 40 and throughout the manuscript, accretion does not equate to NCC. Accretion = NCC – Net Export as per the cited Kleypas et al. (1999) reference. Please correct this throughout the manuscript to avoid mixing NCC and accretion terms before publication.

Validity of the findings

(1) The authors present a great deal of information and cover an impressive range of information in this study; however, I find the lack of data visualization in this manuscript difficult for supporting the conclusions in this manuscript. For example, why weren’t internal and external TA, DIC, temp, sal, DO, and nutrients plotted vs. time for each incubation? This data is important in its own right showing the variability of biogeochemistry in the enclosure and on the reef in general, but is also critical to interpreting the validity of the mixing models proposed in this study. Please show timeseries of the actual data in the manuscript and/or supporting information so that the reader can see how variables changed instead of reading less-clear descriptions of the data throughout the text.

(2) In line 213, why are external O2 and nutrients assumed to be constant? External changes in T, S, TA, and DIC are all clearly discussed during the incubations so I would expect that there would likely also be associated changes in O2 and nutrients outside the incubation tent. In Line 258 after stating that nutrients and O2 are assumed to be constant, the authors state that the ambient concentrations were essentially invariant relative to the in-tent changes but I do not see support of this with the data presented. It seems as though this assumption was due to the lack of measured external O2 data, but this is not clear and I do not see evidence that this is a valid assumption in the context of the rest of the data discussed in this manuscript. Please address this point before publication with either data to support the claims or a statement that this is a speculative, assumption (as are the O2 flux uncertainties) due to the lack of recorded data.

(3) Additionally, much of the accessible data is in the proprietary ‘.mat’ MATLAB format. Please upload this in a more universally accessible format such as ‘.csv’ or similar to improve access for those lacking expensive MATLAB site licenses.

Additional comments

Please see these additional minor comments and suggestions, which I hope will improve the clarity and presentation of the final published manuscript:

Line 34 : “bleaching of reef” to “bleaching of reef corals”

Line 41: “the coral’s overall response to the sum of environmental changes” NCC reflects the community and not the individual coral therefore “coral reef” should be used in place of “coral”

Line 50: There is an expansive literature on quantitative changes in measured depth and accretion from coral cores that would improve the discussion here. For example see the following:

Yates KK, Zawada DG, Smiley NA, Tiling-Range G. Divergence of seafloor elevation and sea level rise in coral reef ecosystems. Biogeosciences. 2017 Mar 15;14(6):1739.

Hubbard DK, Miller AI, Scaturo D. Production and cycling of calcium carbonate in a shelf-edge reef system (St. Croix, US Virgin Islands): applications to the nature of reef systems in the fossil record. Journal of Sedimentary Research. 1990;60(3).

Line 74: What is meant by “calm environments”? Eddy covariance and gradient flux measurements have been successfully employed across a range of reef environments. Please see Takeshita et al. (2016) and references therein for a discussion of these methods:

Takeshita Y, McGillis W, Briggs EM, Carter AL, Donham EM, Martz TR, Price NN, Smith JE. Assessment of net community production and calcification of a coral reef using a boundary layer approach. Journal of Geophysical Research: Oceans. 2016 Aug 1;121(8):5655-71.

Line 77: The authors state “where no accumulation of chemical signals in the water column are expected”, but boundary layers may still allow measurements to be conducted on reef metabolism. Please see Long et al (2013):

Long MH, Berg P, de Beer D, Zieman JC. In situ coral reef oxygen metabolism: An eddy correlation study. Plos One. 2013 Mar 11;8(3):e58581.

Line 82, 83: Consecutive sentences say “Moreover”. Please revise to avoid repetition

Line 174: “two-hourly” is this once every two hours or twice an hour? Please clarify in the text.

Line 180: “immediate” to “immediately”

"CaCO3 dissolution" should be used wherever "dissolution" is written to clarify what is dissolving

Line 347: This implies net calcification, not accretion. Additionally, concentration rates appear to by higher for incubations 3-5, which is likely due to the lack of accounting for leakage in incubations 1 and 2 discussed in line 316.

Line 376: Please clarify what is meant by the following sentence: “The contribution of dissolution is always negative, corresponding to net accretion of CaCO3.”

Line 447: Why compare this 6% calcifier system to the reefs of Bonaire with 16.6% coral cover? Please see Perry et al. (2013) to place the results of this patch reef into a more comparable context:

Perry CT, Murphy GN, Kench PS, Smithers SG, Edinger EN, Steneck RS, Mumby PJ. Caribbean-wide decline in carbonate production threatens coral reef growth. Nature communications. 2013 Jan 29;4:1402.

Line 467: Please see CaCO3 dissolution rates for coral reef sediments around the world to better place these reef sediment dissolution rates in context:

Eyre BD, Cyronak T, Drupp P, De Carlo EH, Sachs JP, Andersson AJ. Coral reefs will transition to net dissolving before end of century. Science. 2018 Feb 23;359(6378):908-11.

Line 467: Middelburg et al. (2005) is not in the reference list. Please ensure that all references are properly documented in the references section.

Line 488: How can we know that 100 grams of fish in the tent is “generous” based on what has been presented in this manuscript?

Paragraph 512: This should additionally discuss the pioneering work by Pawlik and colleagues regarding the role of sponges in POC, DOC, and DIC on coral reefs.

McMurray SE, Stubler AD, Erwin PM, Finelli CM, Pawlik JR. A test of the sponge-loop hypothesis for emergent Caribbean reef sponges. Marine Ecology Progress Series. 2018 Feb 8;588:1-4.

Figure 4: Which incubation number does this figure represent? What do the data from the other incubations look like with respect to the incubation data presented here in figure 4?

The authors declare no funding for this work but surely there were some exceptional amount of costs for both the personnel time, ship time, and resources used in this project. Please state how this project was funded.

Reviewer 2 ·

Basic reporting

No comment

Experimental design

No comment

Validity of the findings

No comment

Additional comments

van Heuven, Webb, et al. use a tent enclosure to estimate metabolic rates of a coral patch at ~20 m depth off Saba, Dutch Caribbean. This is a nice study and would be a useful addition to the literature, particularly because there are very few estimates to date of metabolic rates on reef slopes and it is both surprising and interesting that this reef is net heterotrophic. However, there are a few areas that can be improved with revision, or which need correction.

Title: while this study does report in situ variation in the carbonate system, it seems incidental to the primary purpose of the study, which was to assess metabolic rates of the patch reef. I’d suggest modifying the title to, ‘In situ incubation of a coral patch for assessment of metabolic rates on a reef slope’, or something like that.

First couple sentences of the abstract and first two paragraphs of the introduction: the paper starts with an extensive discussion of anthropogenic stressors on reefs, but this seems out of place as the study really isn’t assessing these impacts. I would substantially reduce if not cut this discussion entirely. Instead, almost all previous studies have investigated reef metabolism on reef flats (typically using a Lagrangian approach) because of the logistical challenges of doing so on a reef slope. This is one of the only studies to measure community metabolism on a reef slope, and I would refocus the introduction in that direction.

Line 40, I would disagree that the accretion/erosion balance of reef communities reflects corals’ overall response to anthropogenic stressors—lots of natural factors impact coral growth rates, certainly not just anthropogenic impacts, and rates of reef accretion are only indirectly related to coral growth rates (for example, reef accretion may be negligible in a wave-swept environment in spite of rapidly growing corals due to periodic physical loss of carbonates).

Line 93, the tent and chemistry measurements were used to assess reef metabolic rates, not the impacts of reef metabolism on in situ carbonate chemistry dynamics.

Line 113, “healthy” as applied to natural systems is often contentious and is problematic as it is rarely defined clearly (what precisely do you mean when you call the reefs “healthy”?). I would avoid the usage entirely. In addition, I’m not sure I would agree that the reefs around Saba are “healthy” even in the context of the wider Caribbean or that the reference cited leads to that conclusion.

Line 186, it doesn’t make sense that insufficient flushing of the umbilical line would lead to lower precision with the one-step titration as compared to the VINDTA since the same procedure was used to collect bottle samples for both. Instead, these levels of precision are about what you’d expect for the two methods.

Line 199 (and throughout), it should be ‘carbonate dissolution’ or ‘dissolution of carbonates’ or something like that, not simply “dissolution”.

Line 266 (and throughout), it should be ‘rates of change of concentrations’ or something like that, not “concentration rates”. Concentrations don’t have rates and written as such it confuses your meaning.

Line 346, the wording is confusing here—you’re using the TA anomaly technique to infer net calcification rates, so TA should be dropping if the reef is net calcifying (or vice versa).

Line 437 (and throughout, especially in tables), I would also provide estimates of metabolic rates in units of mmol/m2/hr as this will allow for easier comparison to other, similar studies and because these values probably translate better to what was measured (i.e., measurements were made over a few hours, not over a year and scaling up short-term fluxes to annual estimates may not be appropriate).

Line 461, ameliorate is misused—I’d just leave it as ‘a complementary tool’.

Line 486-528, in my mind, the net heterotrophy shown by this patch reef (even including net heterotrophy during the daytime!) is perhaps the most surprising and interesting finding of the study. I would consider moving this section to the beginning of the discussion section (i.e., hook the readers with your most interesting result first). Also, how do these rates of net heterotrophy compare to Pnet rates on shallower reefs? This is a neat result and the discussion of this aspect could be expanded a bit to place it in the larger context.

Fig 3, 4 (and throughout) a small thing, but the units for alkalinity should be ueq/kg (microequivalents per kilogram), not umol/kg, and temperature should have units indicated (deg C).


Fig S1, something is way off with the PAR values shown here. Per Table 1, your max values should be 120 umol photons/m2/s during incubation 1, which are sensible values, but here the max values shown are only 2-3 umol photons/m2/s.

---

## Round 0.2 · Major Revisions

Before sending your revision and rebuttal to the reviewers for assessment of the new manuscript and your responses to their suggestions, I ask that you thoroughly revise your rebuttal letter.

As it stands now, there are many instances where you indicate that you have made changes to the text but leave it to the reviewers to find and assess those changes. This places an undue burden on the reviewers, and I ask that you thoroughly re-draft your rebuttal as follows (acknowledging that in some cases you have followed these points but there are some areas where they need work before I return the rebuttal to reviewers):

1) Break each point apart with line and paragraph breaks, indenting your response and using either shading (as you currently have done) or italics/bold or color to distinguish your response from the suggestion.

2) When you change the manuscript text, copy and quote the section changed into the rebuttal, highlighting the changes made. Also include the line numbers in the revised manuscript where these changes are located (as you have generally done).

3) It is my opinion that responses made to reviewers should usually be added to manuscripts. For example, if you provide a lengthy explanation to a reviewer for clarification, either change the text of the manuscript to incorporate some distillation of that explanation (assuming other readers may have similar questions) or be very clear why you are only responding directly to the reviewer and don't feel that you need to change the manuscript accordingly (ie if the reviewer mis-read something or overlooked something). As with all revisions, see point #2 above.

---

## Round 0.3 · Minor Revisions

Both reviewers appreciated your revision and provided additional minor recommendations. Please carefully address these in a subsequent revision.

Reviewer 1 ·

Basic reporting

Overall, the authors have done an excellent job revising the manuscript and responding to this reviewer’s previous comments. My biggest previous concern was regarding the discussion of leak rates and the models constructed. The rebuttal and revisions have largely cleared up these concerns and I believe the resulting manuscript much clearer and easier to follow. I commend the authors again for tackling the difficult problem or addressing leaks in this underwater tent to make such challenging benthic metabolism measurements and in utilizing models to overcome the tent leaks with the stated assumptions, but I also have some further concerns regarding conflicting verbiage in the text in regard to the leaks and the table of ReefBudget data that I believe can be addressed to further improve this manuscript. Please see them below in addition to a few minor comments.

Leaks:
In line 384, they state that this method offers a 1% leak rate—but this would be variable for every attempted seal and this should be stated in the manuscript. The authors state in line 366 they “do not know…that the tent did indeed leak”, but the manuscript clearly states that 1% leak rates were measured in the experiments and that they assume this to be the case for all incubations. The authors later state in their rebuttal that the tent was excessively leaky during two occurrences referencing lines 358. These lines collectively appear to contradict each other…
I commend the authors for maintaining transparency as best as they can about what they know and what they do not know, but these contrasting statements are challenging the core findings of this manuscript and I believe can be revised given a consistent clarifying text to the effect of: (1) 1% leak rates were inferred from 2 of the incubations and assumed to be constant throughout the study (2) 2 of the incubations did appear to have large leaks so those incubation leak rates may not fit the assumed constant 1% leak rate (3) these constant leak rate assumptions and inferred ambient environmental data were necessary given sparse ambient environmental data but future studies can ameliorate these shortcomings by continuous biogeochemical monitoring both within and outside the incubation tents.

Line 217: For the other three incubations, the “(known) variable environment” is actually an assumed variable environment because no external measurements were made. This should be clearly stated in the text.

ReefBudget:
Table S2 is confusing and unclear—I’m assuming that each individual colony is listed separately? I think overall percent cover of each species would improve clarity here. Furthermore, the “% of full planar 4.4 m^2 area” column matches the ReefBudget calcification rates (but please double check units/values!) for most of the coral species so I’m assuming this is in error... The authors state in the methods that there is a total of 4.2% coral cover in the enclosure so perhaps this column is supposed to be growth rates, but then M. faveolata has the growth rate listed for M. franksi (please also note that Montastrea faveolata/franksi/annularis complex is now Orbicella following recent genetic work). This table should list complete genus names for further clarity. The authors state that they followed ReefBudget protocols for this section but the number of errors in this table and lack of described methods makes me weary of the ReefBudget portion of this study. How was ReefBudget conducted? How were % cover and sizes of coral colonies quantified so precisely? Addressing these questions in addition to revising Table S2 will further improve the clarity of this section.

Where is the ambient O2 data reported? Are the authors referring to the 2016 Pelagia O2 vs salinity measurements? But these were collected a full year after the incubations were done and “<5 km away” so some mention of the assumptions associated with this data in the absence of the incubation-contemporaneous ambient measurements should be more clearly stated.

Figure 4e-f: Is the bold red line the complete model and the thinner red line the asymptotic model? This is not clear from the figure labels or caption.

Line 299: The authors state that ambient nutrients were invariable during the incubations, but I do not see actual ambient nutrient data supporting this claim for the duration of the incubations. Where is this coming from?

Please include figure captions and/or labels as appropriate for all of the supplemental materials as I do not see them in the downloaded version.

Experimental design

no further comment

Validity of the findings

no further comment

Additional comments

no further comment

Reviewer 2 ·

Basic reporting

No comment.

Experimental design

No comment.

Validity of the findings

No comment.

Additional comments

The authors have done a very nice job of responding to reviewer comments in their revision and I believe that this version of the manuscript is much stronger thanks to their thoughtful editing. This study provides a useful addition to the literature and I have only a few minor comments which I feel will improve the paper.

Line 18, “Here, we investigate diurnal trends of alkalinity and dissolved organic carbon, oxygen and nutrients…” but also dissolved inorganic chemistry. I’d mention that DIC was monitored as well here in the abstract, for clarity, or perhaps lump alkalinity and DIC together and say “carbonate chemistry, dissolved organic carbon, oxygen, and nutrients”.

Line 27, I’d strongly suggest expressing calcification rates here and in several other places in units of mmol/m2/hr as this makes it easier to compared to NCP fluxes (i.e., same units used) as well as most of the similar data in the literature. It makes sense to scale up to kg/m2/yr when comparing these values to the ReefBudget approach, but in most other cases I think it’s more useful to use the hourly values. In effect, this means that the data will be expressed in both sets of units, depending on the context.

Line 48, NCC reflects responses of the entire community, not just the corals, and this is an important distinction. For example, many corals are not that sensitive to ocean acidification, and some species seem to be totally resistant, whereas dissolution of reef sediment and pavement appears to be much more sensitive to OA. Hence, NCC reflects the overall response of the community, not the responses of the corals (which are themselves only a component of the larger community).

Line 220, 267, 327, “dissolution” should be ‘carbonate dissolution’, ‘CaCO3 dissolution’, or something like that. Be sure to explicit about what is dissolving.

Line 325, I would clarify for the reader that respiration decreases TA slightly *due to the release of nutrients* (though respiration of only carbohydrates/lipids doesn’t impact nutrients or TA).

Line 333-335, “…although exterior AT and CT increased by as much as 60 μmol kg-1 in the latter half of the incubation inside this tent, interior AT and CT during those 4 hours did not increase at rates higher than 2.5 μmol kg-1 hr-1.” The wording is confusing here—you mean that the concentrations in the exterior increased by up to 60 umol/kg during the 4 hr incubation, but concentrations in the interior increased at a max of 2.5 umol/kg/hr?

Line 436-438, “This may be explained by varying community composition such as the absence of macro-bioeroders within our tent, especially not taking into account of mechanical bioerosion caused by parrot fish for instance.” But the chemical technique applied here does not and cannot assess mechanical bioerosion, only net carbonate precipitation and dissolution. This is an important distinction between the techniques—mechanical bioerosion is important to the process of reef accretion, and can be estimated with the ReefBudget approach, but mechanical erosion *cannot* be estimated with a chemical approach, only net carbonate precipitation/dissolution.

Line 439-454, well-written and nice comparison of the pros and cons of these techniques.

---

## Round 0.4 · accepted · Accept

Thank you for your careful attention to the thorough revision process and we expect that the efforts of our reviewers, both of whom are experts in the field, has improved the paper!


#